# The Role of Ranolazine in the Treatment of Ventricular Tachycardia and Atrial Fibrillation: A Narrative Review of the Clinical Evidence

**DOI:** 10.3390/biomedicines12081669

**Published:** 2024-07-26

**Authors:** Kyosuke Murai, Amir Vasigh, Tamás Alexy, Kálmán Tóth, László Czopf

**Affiliations:** 11st Department of Medicine, Medical School, University of Pécs, Ifjúság útja 13, H-7624 Pécs, Hungarytoth.kalman@pte.hu (K.T.); 2Department of Medicine, Division of Cardiology, University of Minnesota, Minneapolis, MN 55127, USA; alexy001@umn.edu; 3Szentágothai János Research Centre, Medical School, University of Pécs, Ifjúság útja 20, H-7624 Pécs, Hungary

**Keywords:** ranolazine, ventricular tachycardia, atrial fibrillation, antiarrhythmic agents

## Abstract

Cardiac arrhythmias are among the leading causes of morbidity and mortality worldwide. While antiarrhythmic drugs traditionally represent the first-line management strategy, their use is often limited by profound proarrhythmic effects. Several studies, including randomized control trials (RCTs), have demonstrated the antiarrhythmic efficacy of ranolazine, which is registered as an antianginal agent, while also establishing its safety profile. This review compiles clinical evidence investigating the antiarrhythmic properties of ranolazine, focusing primarily on ventricular tachycardia (VT) and atrial fibrillation (AF), as they are common rhythm abnormalities with serious complications. Data from RCTs indicate that ranolazine reduces VT incidence, although this effect is not universal. Therefore, we attempt to better describe the patient population that gains the most benefit from ranolazine due to VT suppression. Additionally, ranolazine is known to enhance the conversion rate of AF to sinus rhythm when combined with other antiarrhythmic drugs such as amiodarone, highlighting its synergistic effect in the atrium without provoking ventricular dysrhythmias. Despite the heterogeneity in the currently available data, ranolazine appears to be an effective and safe option for the management of various arrhythmias.

## 1. Introduction

Cardiac arrhythmias not only represent a serious challenge for the field of cardiovascular medicine worldwide but also significantly contribute to morbidity and mortality. Historically, antiarrhythmic drugs represented the therapeutic mainstay to suppress a wide range of arrhythmias. The Cardiac Arrhythmia Suppression Trial (CAST) and subsequent studies showed that, while these drugs are indeed effective in reducing the incidence of some arrhythmias, they may lead to an increased risk of developing others, as well as to sudden cardiac death. This perceived proarrhythmogenic effect led to an overall decline in their utilization and they are now restricted to a limited range of indications [1,2,3]. In contrast, ablation therapy has become an increasingly safer procedure; thus, it has become popular in the management of various arrhythmias, including atrial fibrillation (AF) and ventricular tachycardia (VT). Current therapeutic guidelines recommend catheter ablation in patients with symptomatic AF in whom antiarrhythmic drugs have been ineffective, contraindicated, or not tolerated, or when they are not preferred. In selected patients with symptomatic paroxysmal AF, catheter ablation can also be used as first-line therapy [4,5]. Catheter ablation is an important treatment option for ventricular arrhythmia and in situations when antiarrhythmic drugs are ineffective, not tolerated, or not desired by patients [6]. It is important to note, however, that the high success rate of ablation therapy may be limited to certain clinical scenarios. For instance, it is clearly effective in the management of paroxysmal AF, yet its success rate for the treatment of persistent or permanent AF is lower [7]. Additionally, it is an invasive intervention which carries some risks for complications and requires specialized expertise and costly equipment investments. 

In many clinical scenarios, amiodarone remains a frequently used drug. One of the major drawbacks of amiodarone, however, is its long-term side effect profile: it may adversely affect the structure or function of various organ systems, such as the lungs, liver, thyroid, nervous system, and the skin. The delayed onset of action of the oral formulation is another challenge, particularly when an immediate therapeutic effect is desired [8]. On the other hand, administering an intravenous bolus and continuous infusion may provoke hypotension and worsening cardiogenic shock in select patient populations with severely reduced left or right ventricular function.

Considering these challenges, ranolazine, a piperazine derivative, has been proposed as a potential alternative therapy for arrhythmia management, primarily owing to its excellent and well-established safety profile. It was initially approved by the Food and Drug Administration (FDA) in 2006 for the management of chronic stable angina pectoris. Multiple studies have since demonstrated its antiarrhythmic effect, a property that may present a promising future for this drug. This review was prepared to provide a brief summary on the relevant mechanisms of VT and AF and to summarize the limited clinical data available on the use of ranolazine as an antiarrhythmic agent for the management of these rhythm disorders. Figure 1 presents the main mechanism of action of ranolazine.

VT typically occurs in patients with underlying structural heart disease that most commonly results from myocardial ischemia. However, nonischemic etiologies, such as idiopathic dilated cardiomyopathy, hypertrophic cardiomyopathy, infiltrative diseases including sarcoidosis, congenital and valvular heart disease, may also increase the risk of clinically significant VT. Genetic cardiomyopathies and conduction system disorders, such as long QT and Brugada syndromes, may lead to polymorphic VT in the absence of structural cardiac abnormalities. There are several known mechanisms to induce VT, including triggered activity secondary to early afterdepolarization (EAD) or delayed afterdepolarization (DAD), automaticity, and reentry. EAD occurs during late phase 2 or early phase 3 of the action potential, secondary to its prolonged duration owing to an abnormality in of the following: (1) increased late I_Na_; (2) inward Ca current; (3) sodium–calcium exchange current; (4) decreased outward potassium current. EAD is the trigger for torsade de pointes (TdP). DAD occurs after complete membrane repolarization and is caused by intracellular calcium overload. It may develop in the setting of tachycardia, catecholamine excess, and digitalis toxicity. Given their mechanisms, both EAD and DAD are sensitive to non-dihydropyridine calcium channel blockers, late I_Na_ blockers, and sodium–calcium exchange blockers. Triggered activity can be the underlying cause for sustained VT in the setting of ischemic, dilated, and hypertrophic cardiomyopathies, especially when the arrhythmia originates from a well-defined myocardial territory. Automaticity is a less common cause for focal VT. In the acute phase of MI or during transient ischemia, an increased extracellular potassium concentration leads to partial depolarization in the resting phase and may initiate spontaneous activity. Re-entry due to scarring is the most common mechanism of sustained monomorphic VT in patients with ischemic heart disease. Since re-entry typically occurs at fixed anatomical sites, VT ablation is the optimal treatment option in such cases [6,9,10]. 

There are numerous etiologies of AF, including advanced age, smoking, obesity, alcohol consumption, increased height, hypertension, diabetes mellitus, coronary artery disease, heart failure, valvular disease, and prior cardiac surgery. Other conditions include chronic kidney disease, hyperthyroidism, obstructive sleep apnea, and sepsis owing to the significant catecholamine release or external administration for hypotension management. Genetics may also have an important role. AF often arises from ectopic firing of myocyte sleeves within the pulmonary veins (PVs) draining into of the left atrium. Electrophysiological abnormalities of the PVs are frequently attributed to abnormal shortening of action potential duration and calcium mishandling. A diastolic leak of calcium from sarcoplasmic reticulum enhances inward sodium current via the sodium–calcium exchange, thereby leading to EAD or DAD. The progression of paroxysmal AF to persistent or permanent AF often results from atrial remodeling. Factors contributing to atrial remodeling include stretch-induced fibrosis, hypocontractility, fatty infiltration, inflammation, vascular remodeling, ischemia, ion channel dysfunction, and calcium instability. Aging of the population can shift the primary trigger mechanism for AF, thereby influencing the most appropriate therapeutic interventions. The longer the duration of AF (the greater the AF burden), the more likely its progression to a more permanent form. However, clinical evidence has failed to show a clear association between baseline AF burden and AF-related outcomes such as all-cause mortality and heart failure. Therefore, AF burden cannot be used as a sole factor guiding management [4,5,11,12]. 

In the ensuing sections, we aimed to answer the following primary question: Does ranolazine administration reduce the prevalence or severity of cardiac rhythm disorders, specifically VT and AF, when compared to placebo or the standard of care among patients with a history of arrhythmias or at risk for developing arrhythmias?

## 2. Methods

The authors performed a literature search on the PubMed and Embase databases in October 2023, using the following search terms: “ranolazine AND arrhythmia cardiac”, or “ranolazine AND heart arrhythmia”. All relevant articles published in the English language were carefully examined, specifically those focusing on the primary question detailed above. The scope of this review included observational studies, clinical trials, and randomized controlled studies. Reviews and summaries of molecular studies that did not include human observations were reviewed yet not included when synthesizing and aggregating the data. Additionally, case reports and articles lacking full-text availability were excluded. 

## 3. Results

Following the noted inclusion and exclusion criteria, 11 studies were included in the final synthesized review. Basic study information, the performed interventions, and outcomes are summarized in Table 1 and Table 2. 

### 3.1. Ventricular Tachycardia

The Metabolic Efficiency with Ranolazine for Less Ischemia in Non-ST-Segment Elevation Acute Coronary Syndrome Thrombolysis in Myocardial Infarction 36 (MERLIN-TIMI 36) trial was a randomized, double-blind study in which 6550 patients with NSTEMI were randomly assigned to take either ranolazine or placebo [24]. In a subset analysis, Scirica found ranolazine to significantly reduce the number of VT episodes lasting longer than eight beats during the first 7 days when compared to placebo (166 vs. 265, *p* < 0.001). Continuous ECG monitoring was employed to define arrhythmia burden [13].

Ranolazine in High-Risk Patients with Implanted Cardioverter-Defibrillators (RAID Trial) was another randomized, double-blind, placebo-controlled intention-to-treat study that enrolled 1012 patients with an implantable cardioverter defibrillator (ICD) in situ. Patients were randomly assigned to receive ranolazine or placebo. After a mean of 28.3 months follow-up, ranolazine was found to significantly reduce the incidence of VT and ventricular fibrillation (VF) episodes requiring anti-tachycardia pacing (ATP) or ICD shock, when compared to placebo (433 vs. 650, HR = 0.70 [0.51–0.96]; *p* = 0.028) [14]. In a subset of the RAID trial, Younis observed that the benefits of ranolazine were limited to the following subgroups: (1) patients receiving ranolazine monotherapy (without any concomitant antiarrhythmics); (2) those who have cardiac resynchronization therapy–defibrillator (CRT-D) in place; and (3) patients without atrial fibrillation [15].

### 3.2. Atrial Fibrillation 

Scirica demonstrated in a subset of patients from the MERLIN-TIMI 36 trial that, within the first 7 days after NSTEMI, those in the ranolazine group had a trend towards lower incidence of AF compared to placebo (55 vs. 75, HR = 0.74 [0.52–1.05]; *p* = 0.08) [13]. After one year follow-up, ranolazine led to a substantial decrease in clinically significant AF burden in this population (2.9% vs. 4.1%, HR = 0.71 [0.55–0.92]; *p* = 0.01). Additionally, it reduced the time spent in AF (calculated as proportion of recording time) when compared to placebo (4.4% vs. 16.1%; *p* = 0.015) [16].

Several studies evaluated the potential synergistic effect of ranolazine when used in combination with another antiarrhythmic drug, particularly amiodarone, for the treatment of AF. Koskinas performed a randomized, single-blind study in which 121 patients with recent-onset symptomatic AF were randomly assigned to a combination of amiodarone plus ranolazine or amiodarone alone. Those in the amiodarone plus ranolazine group not only had a significantly higher conversion rate to sinus rhythm within 24 h (53 vs. 42; *p* = 0.024), but the mean time to AF termination was also significantly shorter (10.2 ± 3.3 h vs. 13.3 ± 4.1 h; *p* = 0.001) [17]. Similarly, Tsanaxidis completed a randomized, single-center study enrolling 173 patients with recent-onset AF. Participants were assigned to amiodarone plus ranolazine or amiodarone alone. Amiodarone plus ranolazine significantly increased the conversion rate of AF within 24 h (90 vs. 47; *p* < 0.001) and reduced the mean time to conversion (8.6 ± 2.8 h vs. 19.4 ± 4.4 h; *p* < 0.0001) [18]. 

Simopoulos performed a randomized, single-blind study including 511 patients who developed AF following coronary artery bypass graft surgery (CABG). Patients were randomized to receive amiodarone plus ranolazine or amiodarone only. The combination of ranolazine and amiodarone significantly increased the conversion rate of AF within 24 h (235 vs. 37; *p* < 0.0001) and also reduced the mean time needed for AF termination (10.4 ± 4.5 h vs. 24.3 ± 4.6 h; *p* < 0.0001) [19]. In a retrospective single-center cohort study directly comparing the antiarrhythmic effects of amiodarone and ranolazine in 393 post-CABG patients (within 10–14 days), Miles showed that ranolazine was more effective in reducing the incidence of new-onset AF versus amiodarone (17.5% vs. 26.5%; *p* = 0.035). Importantly, there were significant differences in baseline group characteristics, particularly in the proportion of patients with New York Heart Association (NYHA) class IV symptoms and in ejection fraction (EF). This raises an important and valid concern for potential selection bias [22]. Finally, Hammond published a retrospective cohort study with 76 patients who underwent either CABG or valve surgery and received ranolazine or placebo. After matched-pair analysis, ranolazine was found to significantly reduce the incidence of new onset AF after 7 days when compared to placebo (10.5% vs. 45.6%, OR = 0.09 [0.021–0.387]; *p* < 0.0001) [23]. 

Harmony was a randomized, double-blind, controlled, intention-to-treat study in which 131 patients with paroxysmal AF were randomized into one of five groups: (1) placebo; (2) ranolazine 750 mg; (3) dronedarone 225 mg; (4) dronedarone 150 mg plus ranolazine 750 mg; (5) dronedarone 225 mg plus ranolazine 750 mg. Compared to placebo, the study demonstrated a significantly reduced AF burden when using a combination of dronedarone 225 mg plus ranolazine 750 mg (group 5; 4.8% vs. 11.1%; *p* = 0.008) [20]. 

The Ranolazine in Atrial Fibrillation Following an Electrical Cardioversion (RAFFAELLO) study was a randomized, double-blind, placebo-controlled, multicenter, intention-to-treat study. It enrolled 238 patients with persistent AF (7 days’ to 6 months’ duration) who underwent electrical cardioversion. Patients were randomly assigned to one of four groups: (1) placebo; (2) ranolazine 375 mg; (3) ranolazine 500 mg; (4) ranolazine 750 mg. The treatment duration was 16 weeks. AF recurrence was 56.4%, 56.9%, 41.7%, and 39.7% in groups 1–4, respectively. None of the ranolazine doses extended the time until the first AF recurrence. However, 500 mg ranolazine significantly reduced the rate of AF recurrence versus placebo when patients were still in sinus rhythm after 48 h (HR = 0.56 [0.31–1.01]; *p* = 0.0495) [21].

## 4. Discussion

This narrative review is aimed at compiling existing clinical evidence on the benefits of ranolazine in treating cardiac arrhythmias, specifically VT and AF. 

### 4.1. Brief Mechanism of Action 

Ranolazine is a well-established medication acting as an inhibitor of various ion channels, including late I_Na_, peak I_Na_, the rapid-activating delayed rectifier I_Kr_, and, to a clinically lesser extent, L-type I_Ca_ [25]. The mechanisms through which ranolazine acts on late I_Na_ and peak I_Na_ channels are complex and warrant a limited further discussion, because it will aid in understanding the rationale for the studies reviewed here. In brief, ranolazine is a more effective inhibitor of late I_Na_ channels in ventricular myocytes, whereas it predominantly inhibits peak I_Na_ channels in atrial myocytes [26]. Additionally, I_Kr_ only has a limited role in the atrium. A recent study also demonstrated that ranolazine inhibits TASK-1 potassium channels, which are selectively expressed on the surface of atrial myocytes, although the clinical significance of this inhibition remains limited [27]. 

#### 4.1.1. Late Sodium Current 

Ranolazine functions as an inhibitor of the slowly inactivating domain of the cardiac sodium current, referred to as late I_Na_. Late I_Na_ is found predominantly on the surface of midmyocardial cells (M cells) and in Purkinje fibers [28]. Under physiologic conditions, the amplitude of the late I_Na_ represents less than 1% of the peak I_Na_ current. However, under various pathological conditions, such as myocardial ischemia or heart failure, there is a significant increase in the number of late I_Na_ channels on the surface of ventricular myocytes. The consequently intensified late I_Na_ current prompts an enhanced sodium influx into the cytoplasm of these cells, raising the intracellular concentration of sodium. In response, a reverse-mode sodium–calcium exchange is activated and prompts a rise in cytosolic calcium concentration. In ventricular myocytes, this can impair mechanical relaxation and induce electrical instability by prolonging the cardiac action potential. In addition, it provokes further spontaneous calcium release from the sarcoplasmic reticulum. These events can lead to two different yet closely related activities in the myocardium: (1) transmural dispersion of repolarization (TDR), as indicated by the time between the peak and the end of T wave, and (2) early afterdepolarization (EAD) or T-wave alternans [29]. All of these can potentially induce ventricular arrhythmias, including torsades de pointes [28]. Ranolazine selectively inhibits the late I_Na_ current in ventricular myocytes, thereby limiting sodium overload and cytosolic calcium accumulation. This leads to a decrease in diastolic wall stress and improved coronary flow, thereby promoting enhanced cardiac function [30].

#### 4.1.2. Peak Sodium Current

Ranolazine exploits the differences in peak sodium current between the atrial and ventricular cells. Burashnikov and others demonstrated that ranolazine exerts an atrium-selective, use-dependent inhibition of the maximum action potential slope and prolongs the effective refractory period in isolated canine coronary-perfused atria and ventricles. These results indicate that ranolazine has the potential to treat AF without unwanted proarrhythmic effects at the ventricular level [31,32].

#### 4.1.3. Transmural Dispersion of Repolarization and Early Afterdepolarization

It is known from the literature that ranolazine does not induce EAD or increase TDR. Antzelevitch elegantly mentioned in his study that “dispersion of repolarization secondary to accentuation of electrical heterogeneities intrinsic to ventricular myocardium as the substrate and EADs as the trigger for the development of TdP”. While ranolazine inhibits the I_Kr_ that prolongs the QT interval and would logically trigger TdP, it does not actually lead to TdP, as is the case for other class III antiarrhythmic drugs. Moreover, studies performed on guinea pig and rabbit hearts have shown that ranolazine suppresses TdP induced by selective I_Kr_ blockers. 

The electrophysiological effects of ranolazine were studied in detail by Antzelevich. The ventricular myocardium consists of at least three electrophysiologically distinct cell types: epicardial cells, M cells, and endocardial cells. M cells are characterized by an action potential that is heavily prolonged, compared to that of other ventricular myocardial cell types, in response to pathological conditions and when exposed to certain drugs prolonging the QT interval. This emphasizes normal TDR and results in EAD by reducing the repolarization reserve of the ventricular myocytes. In contrast, ranolazine preferentially prolongs action potential duration in the epicardium but shortens it in M cells, thus leading to a reduction in TDR. This is most likely explained by the inhibitory effect of ranolazine on late I_Na_ [25]. 

### 4.2. Clinical Benefits of Ranolazine

#### 4.2.1. Ventricular Tachycardia

Two large randomized controlled trials have demonstrated the efficacy of ranolazine in reducing the number of VT episodes. Given that the majority of patients in both of these studies had underlying myocardial ischemia (51.0% of the ranolazine group in MERLIN-TIMI 36 had NSTEMI while 47.1% had unstable angina, and 58% in the RAID trial had ischemic cardiomyopathy), the anti-ischemic effects of ranolazine may have contributed to the reduction in the observed arrhythmia burden. In contrast with these results, Benjamin documented a comparable reduction in the incidence of VT with ranolazine in patients with and without myocardial ischemia in the MERLIN-TIMI 36 trial [13]. Similarly, Younis found in the RAID trial that the reduction in VT events did not differ significantly between patients with ischemic and non-ischemic cardiomyopathy [15]. These findings indicate that ranolazine has direct antiarrhythmic properties. 

Our thorough data review suggests that, when used specifically for the management or prevention of VT, certain populations may experience a greater benefit from ranolazine therapy. For example, Younis demonstrated that it was more effective in reducing VT in patients not receiving any other antiarrhythmic agents, individuals without baseline AF, and among those with a CRT-D in situ. These observations are of great importance as, in the RAID trial, 17% of all participants were on an antiarrhythmic agent, with 10% taking amiodarone. Several explanations for these findings may be plausible. Ranolazine may elicit an antiarrhythmic effect similar to that of other agents, and its additional benefit may no further be detectable. Another explanation may be related to its largely rate-independent effect on the ventricle. That is, the antiarrhythmic properties of ranolazine are not enhanced by increased ventricular rates. This suggests that a strong synergistic effect reducing the incidence of VT is highly unlikely to exist between ranolazine and other antiarrhythmic drugs. Interestingly, this is in stark contrast with the results documented in AF, where the combined use of ranolazine and amiodarone demonstrated a strong synergistic effect. Another important observation is that ranolazine appears to have a stronger benefit in patients at low-to-moderate VT risk, and its efficacy is more limited in higher-risk groups (e.g., older individuals, patients with underlying AF, and those already taking antiarrhythmics). Finally, ranolazine was significantly more effective in patients with an implanted CRT-D device compared to those with an ICD. This may be attributed to ranolazine’s mechanism of action which involves the inhibition of late I_Na_ in the ventricle. This makes it especially effective in conditions with greater dispersion in transmyocardial repolarization, such as ischemia and heart failure. Likewise, left bundle branch block (LBBB), a common indication for CRT-D, leads to greater dispersion in repolarization. This indicates a potential synergistic effect explaining the benefit of combining ranolazine with CRT-D therapy [15,33]. Importantly, based on a limited number of animal experiments, ranolazine does not appear to alter the defibrillation threshold and, therefore, it does not reduce the safety margin for successful defibrillation [34]. As it is a frequently used anti-ischemic agent, and ischemic cardiomyopathy is a common indication for primary or secondary prevention ICD implantation, the clinical relevance of this finding might be highly significant. 

#### 4.2.2. Atrial Fibrillation 

Despite a positive trend noted in the MERLIN-TIMI 36 and RAFFAELLO trials, prolonged treatment with ranolazine was not effective in reducing the incidence of new onset and recurrent AF [13,21]. The limited number of studies in this field and their heterogeneity underscore the need for further research in this area. 

Several studies have demonstrated that ranolazine, when combined with amiodarone, promoted a more rapid AF termination and conversion to sinus rhythm. This finding aligns with the experimental data demonstrating a favorable synergy between the two agents that is achieved without any undesirable proarrhythmic effects [30,33]. While discussing the clinical significance of the time reduction until AF termination is beyond the scope of this narrative review, such a decrease may potentially lower hospital costs by shortening inpatient stays. This might ultimately benefit the patients, their caregivers, and society. It is important to emphasize that physicians should always follow their clinical judgment and patient management shall adhere to the most up to date clinical guidelines. Specifically, while the addition of ranolazine to amiodarone favors the maintenance of sinus rhythm, ensuring that no clinically significant myocardial ischemia exists following CABG that would explain arrhythmias is a priority. In addition, optimal medical therapy, including but not limited to aspirin, statins, beta blockers, and a neurohormonal blockade, should be initiated and titrated according to the most recent clinical practice guidelines followed by the addition of amiodarone and ranolazine as necessary [35]. 

### 4.3. Safety Profile of Ranolazine

The risk of new-onset or worsening arrhythmias (proarrhythmic effect) and/or toxicity with prolonged exposure limit the use of many antiarrhythmic drugs currently available on the market. However, substantial clinical evidence supports the safety of chronic ranolazine treatment. It does not cause clinically significant bradycardia or hypotension [36,37]. The most common adverse effects associated with ranolazine use include nausea, constipation, dizziness, fatigue and, less frequently, syncope [14,19,21,24,36,37]. Many clinical trials have established that ranolazine does not provoke TdP or sudden cardiac death, despite its modest QT-prolonging effect [17,38]. This is frequently attributed to the fact that ranolazine does not induce EAD or increase TDR. 

### 4.4. Limitations

This narrative review has certain limitations. First, there is considerable heterogeneity among the studies analyzed. Variations in study design, patient populations, interventions, and outcomes across these may limit the ability to directly compare their results. Second, the small number of studies available on each topic limits our possibility to draw a robust conclusion. Third, this review was limited to VT and AF and did not include other arrhythmias, specifically those based on congenital long QT syndrome and supraventricular tachycardias other than AF.

## 5. Conclusions

Based on available clinical trial data, ranolazine appears to be an effective and safe agent reducing the incidence of VT. However, information regarding its effects on survival is lacking. It may accelerate the conversion of AF into sinus rhythm when used in combination with amiodarone, thus decreasing the time spent in AF and promoting hemodynamic stabilization. The antiarrhythmic properties of ranolazine can provide significant additional clinical benefits to its established anti-ischemic and antianginal effects. 

There is a growing number of studies on the use of ranolazine in cardiovascular diseases beyond angina pectoris, including arrhythmias and heart failure. Its mechanism of action is complex and not yet fully understood. Since ranolazine was introduced to the U.S. market in 2006, its long-term safety and efficacy are not yet well-established, which may hinder its use as a first-line agent. Gathering further clinical evidence through clinical trials and observational studies is essential to solidify the information on the beneficial cardiovascular effects of ranolazine. 

Although the term “proarrhythmic” is commonly used in the context of antiarrhythmic therapy, whether the full scale of its impact is understood remains unclear. Further biomolecular studies are essential to understand the underlying mechanisms which may, in turn, assist in choosing the most appropriate antiarrhythmic strategy for individual patients. 

## Figures and Tables

**Figure 1 biomedicines-12-01669-f001:**
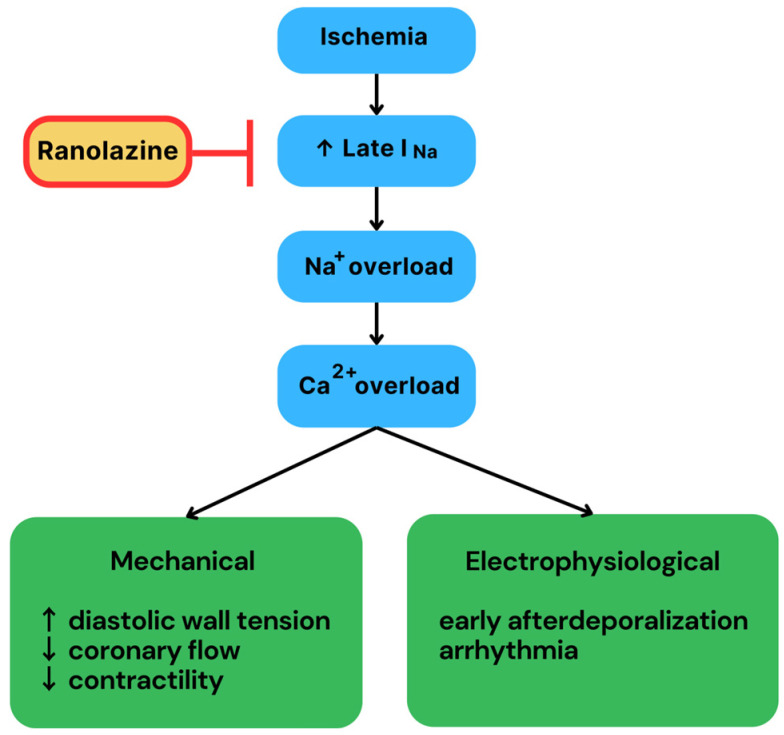
Summary of the most clinically important effects of ranolazine.

**Table 1 biomedicines-12-01669-t001:** Effect of ranolazine on reducing VT burden or severity.

Authors	Population	Intervention/Comparison	Outcome
Scirica (MERLIN-TIMI 36) [13]	Randomized, double-blind study; 6351 patients with NSTEMI or unstable angina randomly assigned to ranolazine (63 ± 11 years, 2093 males) or placebo (63 ± 11 years, 2031 males)	IV ranolazine with 200 mg bolus then 80 mg/h infusion for 12–96 h, then PO 1000 mg ranolazine BID. Continuous ECG monitoring for the first 7 days	The number of VT episodes exceeding 8 beats was significantly reduced in the ranolazine group vs. placebo (166 [5.3%] vs. 265 [8.3%]; *p* < 0.001)
Zareba, Younis (RAID trial) [14,15]	Randomized, double-blind, placebo-controlled, multicenter, intention-to-treat study; 1012 patients with ICD randomly divided into ranolazine (64.3 ± 10.3 years, 410 males) and placebo (64.2 ± 9.9 years, 416 males) groups	Ranolazine 500 mg BID for 1 week, increased to 1000 mg BID if dosage tolerated. Follow-up for mean 28.3 ± 15.8 months	Significantly reduced recurrence of VT/VF requiring ATP or ICD shock in the ranolazine group vs. placebo (433 vs. 650, HR = 0.70 [0.51–0.96]; *p* = 0.028). The benefit of ranolazine was limited to the following subgroups: (1) ranolazine monotherapy (HR = 0.68 [0.55–0.84] vs. HR = 1.33 [0.90–1.96], *p* = 0.003); (2) implanted CRT-D (HR = 0.64 [0.47–0.86] vs. HR = 0.94 [0.74–1.18]; *p* = 0.047); (3) no AF (HR = 0.66 [0.54–0.81] vs. HR = 1.56 [1.02–2.39], *p* = 0.003)

Abbreviations: NSTEMI: non-ST-elevation myocardial infarction; VT: ventricular tachycardia; ICD: implantable cardioverter defibrillator; BID: twice daily; VF: ventricular fibrillation; ATP: anti-tachycardia pacing; CRT-D: cardiac resynchronization therapy–defibrillator; AF: atrial fibrillation; IV: intravenous; PO: per os; HR: hazards ratio; mg: milligram.

**Table 2 biomedicines-12-01669-t002:** Effect of ranolazine on AF prevention and on AF conversion to sinus rhythm.

Authors	Population	Intervention/Comparison	Outcome
Scirica (MERLIN-TIMI 36) [13,16]	Randomized, double-blind study; 6351 patients with NSTEMI or unstable angina randomly assigned to ranolazine (63 ± 11 years, 2093 males) or placebo (63 ± 11 years, 2031 males)	IV ranolazine 200 mg bolus followed by 80 mg/h infusion for 12–96 h, then 1000 mg PO BID. Continuous ECG monitoring for the first 7 days, follow-up for 12 months	Trend towards reduced AF occurrence within the first 7 days with ranolazine vs. placebo (55 [1.7%] vs. 75 [2.4%], HR = 0.74 [0.52–1.05]; *p* = 0.08). Reduced clinically significant AF * after 12 months in ranolazine group vs. placebo (2.9% vs. 4.1%, HR = 0.71 [0.55–0.92]; *p* = 0.01)
Koskinas [17]	Randomized, single-blind study; 121 patients with symptomatic AF (<48 h) randomly assigned to amiodarone plus ranolazine (66 ± 11 years, 25 males) or amiodarone-only (64 ± 9 years, 29 males)	Amiodarone: 5 mg/kg IV loading dose in 1 h then 50 mg/h for 24 h or until cardioversion Ranolazine: 1500 mg PO at randomization	Amiodarone plus ranolazine significantly increased AF conversion rate within 24 h (53 [87%] vs. 42 [70%]; *p* = 0.024) and reduced mean time to AF conversion (10.2 ± 3.3 vs.13.3 ± 4.1 h; *p* = 0.001)
Tsanaxidis [18]	Randomized, single-center study; 173 patients with recent-onset AF randomly assigned to amiodarone plus ranolazine (92, 70 ± 10 years, 38 males) and amiodarone-only (81, 67 ± 11 years, 41 males)	Amiodarone: 5 mg/kg IV loading dose, 50 mg/h maintenance infusion. After conversion, 200 mg PO BID for a week and then 200 mg PO daily for a week. Ranolazine: 1000 mg POthen 375 mg BID 6 h after arrhythmia termination	Amiodarone plus ranolazine significantly increased conversion of AF within 24 h (90 [98%] vs. 47 [58%]; *p* < 0.001) and reduced mean time to AF termination (8.6 ± 2.8 h vs. 19.4 ± 4.4 h; *p* < 0.0001) vs. the amiodarone-only group
Simopoulos [19]	Randomized, single-blind study; 511 patients with post-CABG AF randomly assigned to amiodarone plus ranolazine (65.3 ± 9.5 years) or amiodarone-only (65.5 ± 9.6 years)	Amiodarone: 300 mg IV in 30 min, followed by 1125 mg in 36 h Ranolazine: 500 mg PO, 375 mg after 6 h and then 375 mg PO BID	Amiodarone plus ranolazine increased AF conversion to sinus rhythm significantly within 24 h (235 [91.8%] vs. 37 [14.5%]; *p* < 0.0001) and reduced mean time of AF conversion (10.4 ± 4.5 h vs. 24.3 ± 4.6 h; *p* < 0.0001) vs. amiodarone alone
Reiffel (Harmony trial) [20]	Randomized, double-blind, placebo-controlled, intention-to-treat study; 131 patients with paroxysmal AF randomized to (1) placebo (72 ± 8.4 years, 13 males); (2) ranolazine 750 mg (70 ± 10.8 years, 10 males); (3) dronedarone 225 mg (75 ± 7.8 years, 10 males); (4) dronedarone 150 mg plus ranolazine 750 mg (73 ± 9.4 years, 15 males); (5) dronedarone 225 mg plus ranolazine 750 mg (71 ± 7.1 years, 15 males)	Each given BID for 12 weeks	Significantly reduced AF burden ** in group 5 vs. placebo (4.8% vs. 11.1%; *p* = 0.008)
De Ferrari (Raffaello trial) [21]	Randomized, double-blind, placebo-controlled, multicenter, intention-to-treat study; 238 patients with persistent AF (7 days to 6 months) randomly assigned to: (1) placebo (55, 65.2 ± 9.5 years, 41 males); (2) ranolazine 375 mg (65, 66.9 ± 11.8 years, 46 males); (3) ranolazine 500 mg (60, 65.5 ± 8.5 years, 51 males); (4) ranolazine 750 mg (58, 63.6 ± 11.3 years, 46 males)	Each given BID for 16 weeks after electrical cardioversion	Significantly reduced AF recurrence in group 3 vs. placebo when patients were still in sinus rhythm after 48 h (HR = 0.56, [0.31–1.01]; *p* = 0.0495)
Miles [22]	Retrospective, single-center cohort study; 393 patients after CABG received either amiodarone (211, 64.9 ± 10.9 years, 162 males) or ranolazine (182, 66.7 ± 9.3, 127 males)	Amiodarone: 400 mg daily preoperatively (7 days prior to elective CABG or immediately before urgent CABG), 200 mg BID postoperatively for 10–14 days Ranolazine: 1500 mg on the day prior to elective CABG or on the day of urgent CABG. 1000 mg BID postoperatively for 10–14 days	Ranolazine significantly reduced the incidence of new-onset AF compared to amiodarone (17.5% vs. 26.5%; *p* = 0.035)
Hammond [23]	Retrospective, single-center cohort study; after matched-pair analysis, 114 patients post-CABG or valve surgery received either ranolazine (57, 60.3 ± 11.1 years, 38 males) or placebo (57, 59.6 ± 11.5 years, 38 males)	Preoperative ranolazine 1000 mg on the morning of surgery, 1000 mg BID afterwards for 7 days	Ranolazine significantly reduced the incidence of new-onset AF vs. placebo (10.5% vs. 45.6%; OR = 0.09 [0.021–0.387]; *p* < 0.0001)

Abbreviation: NSTEMI: non-ST-elevation myocardial infarction; BID: twice daily; HR: hazard ratio; CABG: coronary artery bypass graft; OR: odds ratio. * Clinically significant AF as used by the authors includes paroxysmal AF (AF burden between 0.01% and 98%) and predominantly chronic AF (AF burden more than 98%). ** AF burden was calculated as the proportion of recording time (%) in AF.

## Data Availability

Not applicable.

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
