# Peer review of "The Role of Ranolazine in the Treatment of Ventricular Tachycardia and Atrial Fibrillation: A Narrative Review of the Clinical Evidence"

_biomedicines, 2024, doi:10.3390/biomedicines12081669_

Round 1

Reviewer 1 Report

Comments and Suggestions for Authors

Ranolazine is not a new drug, its action is pleiotropic and includes primarily metabolic intervention but also antiarrhythmic effects resulting from modulation of ion channel conductivity. While reading the works, I unfortunately had the impression that the authors were in a hurry to reach the conclusion, and therefore both the introduction and the discussion remained as if they were not fully developed.

This topic is important, there are no works summarizing the effectiveness of ranolazine and this is a huge advantage of this work. In order to improve the quality of the work, the authors need to correct several elements of the manuscript.

- the introduction is chaotic. lacks a systematic approach. Please divide the arrhythmias, isolate the mechanisms, indicate common mechanisms for different arrhythmias. (maybe figure 1)

- the drug itself - mechanisms of action, even if briefly, could be enriched with a figure (Figure 2)

- please discuss the mechanisms of action of ranolazine in terms of the mechanisms of the arrhythmias in question.

- Please discuss the problem of AF development in accordance with the agreed key, taking into account different age populations. After all, a different mechanism leads to AF in a marathon runner or rower than in an elderly person.

- please discuss the mechanisms of VT and classify them. Discuss the causes of storage diseases, ischemic, post-COVID-19 or other inflammations.

- in the introduction, the authors treat ablation as an almost optional method, and there are specific indications (please use the current guidelines).

- please pay attention to the average age of the population and therapeutic interventions or clinical settings. Middle-aged patients from 60 to 80, so probably with significant atherosclerosis, after heart attacks (there was definitely significant ischemia here) or CABG... In these patients, the course of action is clearly defined and appropriate diagnostics and invasive treatment must be carried out and only then consider ranolazine. Authors write relatively a lot about amiodarone, but on the other hand, the reason for the presence of amiodarone was the primary cause of the problem... ischemia. This type of discussion must be conducted without becoming fixated on using only one drug in all circumstances.

- please place the table rotated 90 degrees.

Author Response

We would like to thank the Reviewer for your time thoroughly reviewing our manuscript and providing excellent feedback and suggestions. Please find below our point-by-point answers and the modifications included in the most up to date version of the manuscript (indicated in red).

Comment 1: The introduction is chaotic, lacks a systematic approach. Please divide the arrhythmias, isolate the mechanisms, indicate common mechanisms for different arrhythmias.

Response 1: The introduction section was modified and reorganized to make it more understandable and easier to follow. To provide a context to the antiarrhythmic use of ranolazine in VT and AF, general arrhythmia mechanisms were described. In addition, the mechanism of VT and AF were summarized along with the reasoning for selecting these two distinct arrhythmias for the review. [Page 1-3]

Comment 2: The drug itself- mechanisms of action, even if briefly, could be enriched with a figure.

Response 2: Thank you for this comment. We added Figure 1 to the manuscript depicting the primary proposed antiarrhythmic mechanism of ranolazine. [Page 2]

Comment 3: Please discuss the mechanisms of action of ranolazine in terms of the mechanisms of the arrhythmias in question.

Response 3: Thank you fro the suggestion, this is indeed and important point. We expanded the discussion on the mechanism of action of ranolazine. It now details ranolazine's electrophysiological effects on transmural dispersion of repolarization and early afterdepolarization. Both of these effects are critically important in the development of VT and AF. [Page7-8, Paragraph 4.1.3]

Comment 4: Please discuss the problem of AF development in accordance with the agreed key, taking into account different age populations. After all, a different mechanism leads to AF in a marathon runner or rower than in an elderly person.

Response 4: Thank you for the suggestion. We now describe the most common AF etiologies in the introduction. Additionally, we discuss aging as an important factor influencing the pathomechanism and treatment of AF. Chaotic microreentries often originate from around the pulmonary veins frequently promote the development of AF. However, other underlying mechanisms such as mechanisms such as myocardial ischemia, myocardial scar formation, inflammation, deposition of certain metabolites in the myocardium, atrial dilation, and strain of the atrial wall may also play an important role. These show a clear age-association, possibly providing a shift in the evoking factors and influencing the therapeutic target mechanisms with advancing age. [Page 2]

Comment 5: Please discuss the mechanisms of VT and classify them. Discuss the causes of storage diseases, ischemic, post-COVID-19 or other inflammation.

Response 5: The most common etiologies and VT mechanisms are now included in the introduction. The reviewer is asked to also kindly refer to Comment #1. [Page 2]

Comment 6: In the introduction. the authors treat ablation as an almost optional method, and there are specific indications (please use the current guidelines).

Response 6: The introduction has been updated and now we also reference the current guidelines for the indications for ablation therapy. [Page 1, Paragraph 1, Line 8-13]

Response 7: Please pay attention to the average age of the population and therapeutic interventions or clinical settings. Middle-aged patients from 60 to 80, so probably with significant atherosclerosis, after heart attacks (there was definitely significant ischemia here) or CABG... In these patients, the course of action is clearly defined, and appropriate diagnostics and invasive treatment must be carried out and only then consider ranolazine. Authors write relatively a lot about amiodarone, but on the other hand, the reason for the presence of amiodarone was the primary cause of the problem... ischemia. This type of discussion must be conducted without becoming fixated on using only one drug in all circumstances.

Response 7: Thank you for this comment. We now emphasize in the manuscript the importance of comprehensive clinical assessment and management in the postoperative care of CABG patients, prioritizing first line pharmacological therapies before considering more complex strategies. This includes the prevention and treatment of associated arrhythmias. [Page 8,9, Paragraph 4.2.2]

As the reviewer pointed out, management of the underlying pathological condition, e.g., ischemia, is always the utmost priority. Note however, that since ranolazine seemed to be equally effective in ischemic and in non-ischemic cardiac disease reducing the risk of arrhythmias, we did not consider the antiarrhythmic efficacy of ranolazine to be a consequence of anti-ischemic properties. [Page 8. Paragraph 4.2.1]. While amiodarone is commonly used and is highly effective in the setting of postoperative AF in patients after CABG, its effect of onset may not be immediate.

Comment 8: Please place the table rotated 90 degrees.

Response 8: Thank you for the suggestion. We plan to discuss with the editorial team the most optimal table layout for visualization purposes. We will be glad to rotate if it provides a more optimized fit.

Again, we would like to thank the Reviewer's time and effort. Incorporating the recommended changes and additions certainly improved the paper.

Reviewer 2 Report

Comments and Suggestions for Authors

the scientific value of the article is low.

very old references were analyzed.

the elements of novelty and practical implications are missing

Comments on the Quality of English Language

Moderate editing of English language required

Author Response

Thank you very much for taking time and carefully reading our manuscript; we appreciate your feedback. Please find our answers to your remarks below.

Comment 1: The scientific value of the article is low.

Comment 1: We performed a careful literature review on the antiarrhythmic efficacy in an era when it is not yet registered for this particular indication. We considered performing a systematic review, but the available clinical trials lacked the scrutiny and data allowing us to perform this successfully. Therefore, we decided to summarize the most relevant articles in a form of a narrative review. Our hope is to draw clinician attention to a potentially beneficial effect of a classic and approved cardiovascular drug in an era of disappointments with novel antiarrhytmic agents.

Comment 2: Very old references were analyzed.

Response 2: The above-described clinician skepticism towards novel antiarrhythmic agents has limited the number of novel clinical trials with antiarrhythmic agents in recent years. We referenced trials and manuscripts that we found to be the most recent in the filed.

Comment 3: The elements of novelty and practical implications are missing.

Response 3: Only sporadic reports exist around the antiarrhythmic effects of ranolazine and it is not incorporated into the recent clinical guidelines. Therefore, clinician familiarity with this potentially beneficial effect is limited. Therefore, we believe such literature review will be beneficial to the field.

Reviewer 3 Report

Comments and Suggestions for Authors

The authors argued that while antiarrhythmic drugs are the primary treatment, their use is limited by proarrhythmic effects. Ranolazine, initially an antianginal agent, has shown antiarrhythmic efficacy and a good safety profile in several studies, including RCTs. This review focuses on ranolazine's effects on ventricular tachycardia (VT) and atrial fibrillation (AF). RCTs indicate that ranolazine reduces VT incidence, though not universally, and enhances AF conversion to sinus rhythm, especially when combined with drugs like amiodarone. Despite data variability, ranolazine is a promising, safe option for managing arrhythmias.

General comments

This is a manuscript addressing “The Role of Ranolazine in the Treatment of Ventricular Tachycardia and Atrial Fibrillation: A Narrative Review of the Clinical Evidence”.  The reviewer has some concerns need to be addressed.

1)      The following conclusion is not based on the results: “especially in cases when left ventricular ejection fraction is severely reduced.”

2)      Table 2: The percentages are necessary after the numbers in the “outcome” column. Additionally, the results of Simopoulos appear to be incorrect (235 vs. 37). The text should also be revised accordingly.

3)      It is difficult to review without page or line numbers.

4)      "M cells" should be more specifically described.

Author Response

We would like to thank the reviewer for your time reviewing our paper and providing constructive feedback. Please find our point by point responses to the comments below; we have incorporated all changes in the manuscript. These changes are indicated in blue color.

Comment 1: The following conclusion is not based on the results: "especially in cases when left ventricular ejection fraction is severely reduced."

Response 1: Thank you for this recommendation. We agree with the suggestion completely and have removed the sentence from the most recent version of the manuscript.

Comment 2: Table 2: The percentages are necessary after the numbers in the "outcome" column. Additionally, the results of SImopoulos appear to be incorrect (235 vs. 37). The text should also be revised accordingly.

Response 2: Appreciate this suggestion. We have now included the percentages in the "outcome" column of the Table. The results of Simopoulos et al. (235 vs. 37) have been thoroughly reviewed by the authors and were found to be correct as per the cited manuscript. We will be glad to revisit if you can indicate the specific concern.

Comment 3: It is difficult to review without page or line numbers.

Response 3: Thank you for this suggestion. We completely agree with this concern however we utilized the required template by the Journal's article submission system.

Comment 4: "M cells" should be more specifically described.

Thank you for this suggestion. In the updated version of the paper, we specifically describe "M cells" as midmyocardial cells when they first appear in the manuscript. [Page 7, Paragraph 4.1.1, Line 2]

Again, we would like to thank the Reviewer's time and effort. Incorporating the recommended changes and additions certainly improved the paper.

Round 2

Reviewer 1 Report

Comments and Suggestions for Authors

The authors have significantly modified the manuscript. I believe it may be considered for publication

Author Response

We would like to thank you for the time reviewing our manuscript and providing feedback. We agree that there were immensely helpful to further improve the value and quality of the paper.

Reviewer 2 mentioned that minor modifications in English language may be necessary. We thank the reviewer for this comment and indeed we have reviewed the paper again and made minor modifications to the wording and some sentence structure. Please note that none of these enhancements affected the content of the manuscript.

We trust that with these modifications the paper will be acceptable for publication.

Reviewer 2 Report

Comments and Suggestions for Authors

the article was improved

I agree to be published

Comments on the Quality of English Language

Moderate editing of English language required

Author Response

We would like to thank you for the time reviewing our manuscript and providing feedback.

Comment 1: Moderate editing of English language required

Response 1: We thank the reviewer for this comment and indeed we have reviewed the paper again and made minor modifications to the wording and some sentence structure. Please note that none of these enhancements affected the content of the manuscript.

We trust that with these modifications the paper will be acceptable for publication.

Reviewer 3 Report

Comments and Suggestions for Authors

The authors have corrected the manuscript according to the reviewers' comments.

Author Response

We would like to thank you for the time reviewing our manuscript and providing feedback. We agree that these were immensely helpful to further improve the value and quality of the paper.

Reviewer 2 mentioned that minor modifications in English language may be necessary. We thank the reviewer for this comment and indeed we have reviewed the paper again and made minor modifications to the wording and some sentence structure. Please note that none of these enhancements affected the content of the manuscript.

We trust that with these modifications the paper will be acceptable for publication.